# Genetic Diversity and Structure of Korean Pacific Oyster (*Crassostrea gigas*) for Determining Selective Breeding Groups

**DOI:** 10.3390/biology14040449

**Published:** 2025-04-21

**Authors:** Kang-Rae Kim, Dain Lee, Kyung-Hee Kim, Hyun Chul Kim, So Hee Kim, Su Jin Park, Deok-Chan Lee

**Affiliations:** 1Southeast Sea Fisheries Research Institute, National Institute of Fisheries Science, Namhae 52440, Republic of Korea; kimkangrae9586@gmail.com (K.-R.K.); thgml1327@korea.kr (S.H.K.); ssujini@korea.kr (S.J.P.); 2Fish Genetics and Breeding Research Center, National Institute of Fisheries Science, Geoje 53334, Republic of Korea; gene419@korea.kr (D.L.); kyunghee@gmail.com (K.-H.K.); hckimgnu@korea.kr (H.C.K.)

**Keywords:** microsatellite, effective population size, genetic structure, genetic diversity, *Crassostrea gigas*

## Abstract

This study analyzed the genetic diversity and structure of thirteen wild populations of the Pacific oyster (*Crassostrea gigas*) collected from Korea. The findings revealed overall low genetic diversity across populations and suggested that all populations belonged to a single genetic group. Based on these results, potential reference populations were identified for future selective breeding programs aimed at improving the genetic diversity of *C. gigas* in Korea. This research provides valuable baseline information to support sustainable oyster breeding strategies.

## 1. Introduction

*Crassostrea gigas* (Pacific oyster) is a key species in the Korean aquaculture industry [1]. In recent years, however, the production of farmed *C. gigas* has shown a marked decline, raising concerns about the long-term sustainability of oyster farming. One of the primary factors contributing to this decline is the reduction in genetic diversity within cultured populations [2,3]. A decrease in genetic diversity has been reported to correlate with lower early survival rates, particularly during the larval stages [4]. As a result, significant efforts have been made to develop strategies aimed at preserving and enhancing genetic variation in aquaculture stocks [3,4].

Among these strategies, selective breeding approaches that explicitly consider both the number and diversity of alleles are recognized as crucial for restoring and maintaining genetic diversity [5]. Increasing the frequency of heterozygous individuals carrying diverse allelic combinations can improve the adaptive potential and overall genetic health of oyster populations. Therefore, the development of well-structured breeding designs and targeted selection programs is essential for enhancing genetic diversity and ensuring the long-term viability of the Pacific oyster aquaculture industry [6]. Meanwhile, the genetic diversity of wild Pacific oysters from Korea has been shown to be high [3].

Previous studies in *C. gigas* have reported that the observed heterozygosity (*H*_O_) is not similar to the expected heterozygosity (*H*_E_) but rather is reduced [3]. A decrease in *H*_O_ compared to *H*_E_ is reported to indicate lower genetic diversity because less heterozygosity is observed [3]. Additionally, since fixed traits are preferred in the aquaculture industry, there is a problem that fixed traits have low genetic diversity [7]. Therefore, there is a need to investigate larger numbers of wild Pacific oyster populations to improve the genetic diversity of farmed varieties. Wild Pacific oysters are suitable as a reference population for increasing and improving genetic diversity. Because the genetic diversity of Pacific oysters in culture has already decreased and rare alleles have been lost, it is important to introduce populations with high genetic diversity from wild populations [8].

The Ministry of Oceans and Fisheries in Korea has established legal guidelines for breeding Pacific oysters (https://www.mof.go.kr/doc/ko/selectDoc.do?bbsSeq=35&docSeq=53904&menuSeq=885, accessed on 18 February 2025). Therefore, in order to breed and improve Pacific oysters as varieties according to these laws, securing populations with genetic diversity is the first step [3,8]. Second, it is important to confirm the genetic structure of the population in order to select a reference population [9,10,11]. If genetic differentiation is high among populations, it is not suitable as a reference population to be used as a variety [12,13]. The reference population should be a population with similar genetic background, and selection of genetically differentiated populations is not appropriate because bias occurs when a population with different genetic backgrounds is introduced in breeding [14].

In addition to considering genetic diversity and structure in selecting a reference population, the effective population size should be considered [15]. Since the effective population size maintains the genetic diversity of the population and is related to adaptability, it is very important to increase genetic diversity and adaptability for the purpose of breeding improvement [15].

In this study, we investigated the genetic diversity of 13 wild populations in Korea to ultimately select wild populations for improvement by increasing genetic diversity. In addition, we focused on selecting a reference population to increase genetic diversity by checking the effective population size and the presence of a bottleneck phenomenon and considering genetic diversity and effective population size. The genetic structure investigation between populations was conducted to check whether the genetic background was similar or whether there was a difference in differentiation, and the identified genetic differences were considered in selecting the reference population. The genetic diversity and genetic structure investigation from the wild population of *C. gigas* in Korea can be used as basic data for improving varieties in Pacific oysters.

## 2. Materials and Methods

### 2.1. DNA Extraction and Sampling

*C. gigas* serves as an inland fishery resource for both commercial fishing and aquaculture in Korea and is exempt from animal ethics approval, classified under a minimal suffering level of A. For this study, samples were collected from thirteen wild populations of *C. gigas*, with specific locations and their latitudes and longitudes shown in Figure 1 and Table 1. Sampling took place in 2024, and mantle tissue from each Pacific oyster was collected and preserved in 99% ethanol. Genomic DNA (gDNA) was then extracted using the DNeasy Blood & Tissue Kit (QIAGEN, Hilden, Germany) following the manufacturer’s protocol. The extracted genomic DNA was diluted to a concentration of 50 ng/μL and stored at −20 °C to prepare for microsatellite loci amplification.

### 2.2. Microsatellite Loci Genotyping Analysis

Fifteen microsatellite loci, previously developed in past studies (CG4, CG11, CG18, CG18N, CG22-2, CG22-5, CG22-7, CG22-17, CG22-32, CG22-59, CG22-67, CG23N, CG36, ucdCg109, and ucdCg175), were selected for PCR analysis, which was performed using the Applied Biosystems™ ProFlex™ PCR System (Thermo Fisher Scientific, Foster City, CA, USA) [16,17]. The PCR reaction mix contained 50 ng of genomic DNA, 0.5 units of AccuPower^®^ PCR PreMix (Bioneer, Daejeon, Republic of Korea), 0.5 μM of a fluorescent dye-labeled forward primer (FAM, HEX, TAMRA, and ATTO565), and 0.5 μM of a reverse primer. The PCR cycling conditions included an initial denaturation at 95 °C for 5 min, followed by 35 cycles of 95 °C for 20 s, 60 °C for 20 s, and 72 °C for 20 s. A final extension step was held at 72 °C for 10 min, then maintained at 8 °C. To confirm the presence and size of amplified fragments, each PCR product was electrophoresed on a 2% agarose gel. For allele sizing, PCR products were prepared with GeneScan™ 500 ROX size standard (Thermo Fisher Scientific) and HiDi™ formamide, then denatured at 95 °C for 2 min, followed by cooling at 4 °C. The allele sizes were measured using an Applied Biosystems™ ABI 3730xl DNA Analyzer (Thermo Fisher Scientific), and genotyping was conducted using GeneMapper software (version 5) [18].

### 2.3. Genetic Diversity Analysis

To evaluate scoring errors in microsatellite loci, MICRO-CHECKER software (version 2.2.3) was used [19]. Genetic diversity was assessed by determining the number of alleles (*N*_A_), expected heterozygosity (*H*_E_), and observed heterozygosity (*H*_O_) via CERVUS software (version 3.0) [20]. The inbreeding coefficient (*F*_IS_) and Hardy–Weinberg equilibrium (*P*_HWE_) tests were conducted with GENEPOP (version 4.2) [21] and ARLEQUIN (version 3.5) [22]. Population bottlenecks were investigated using two approaches: first, the BOTTLENECK software (version 1.2.02) [23] examined heterozygote excess based on the infinite alleles mutation model (IAM) [24], while the second approach applied a two-phase mutation model (TPM) and stepwise mutation model (SMM) [24] using parameters of 10% variance for TPM and 90% SMM. Both methods ran for 10,000 iterations, with significance tested via the Wilcoxon signed-rank test [25]. The effective population size (*N*_e_) was estimated using the linkage disequilibrium approach in NeEstimator software (version 2.1) [26]. The M-ratio value for estimating historical bottlenecks was calculated using Arlequin (version 3.5) [22].

### 2.4. Genetic Structure of Populations Analysis

Genetic structure clustering analysis was performed using STRUCTURE software (version 2.3) [27], applying a Bayesian model approach. To identify the most suitable number of populations (*K*), values from 1 to 10 were tested, with an admixture model to account for the mixed water systems. The analysis involved a burn-in period of 10,000 iterations, followed by 100,000 iterations of Markov chain Monte Carlo, repeated 10 times for accuracy. The optimal number of clusters was then determined by analyzing results for each *K* value using STRUCTURE SELECTOR (beta version) [28]. Additionally, a discriminant analysis of principal components (DAPC) was conducted on the microsatellite dataset using the ADEGENET package (version 2.1.3) [29] in R, a non-Bayesian clustering approach.

## 3. Results

### 3.1. Genetic Diversityof Wild Populations

Genetic diversity was analyzed, and inbreeding coefficients were measured across 13 populations (Table 2). The range of average allele numbers was 7.60–12.27, with *H*_O_ ranging from 0.367 (lowest in UD) to 0.418 (highest in SA). *H*_E_ ranged from 0.580 to 0.661. *F*_IS_ values ranged from 0.297 to 0.417, with SA showing the lowest values and JD the highest. All populations showed deviations from Hardy–Weinberg equilibrium (HWE).

### 3.2. Bottleneck Test and Effective Population Size Analysis

The bottleneck test results for the 13 populations showed that all populations had values below 0.05 under the IAM model, indicating a bottleneck effect (Table 3). The mode-shift test displayed an L-shape, suggesting a normal allele frequency distribution. The M ratio was below 0.6 in all populations. Excluding WD and YD, where effective population size (*N*_e_) could not be measured, *N*_e_ values ranged from 28 to 2491, with JD having the largest *N*_e_ and AD the smallest.

### 3.3. Genetic Structure and Genetic Differentiation of Wild Populations

The *F*_ST_ range, indicating genetic differentiation, was from 0.001 to 0.057 (Table 4). The genetic structure of the 13 populations showed the most appropriate delta *K* value at *K* = 2 in the STRUCTURE analysis (Figure 2). Additionally, a high delta *K* value was also observed at *K* = 3. The genetic structure of the population estimated by DAPC was one group: AD, CC, CSD, GGD, JD, MA, SA, SJ, SR, UD, WD, WSD, and YD (Figure 3). Based on the DAPC results, the AMOVA analysis showed 1.22% significant variance among groups and 64.70% within populations (Table 5).

## 4. Discussion

### 4.1. Genetic Diversity of Wild Populations

Investigating the genetic diversity of Korean *C. gigas* populations is an essential initial step in improving the species through selective breeding. Thus, the genetic diversity of thirteen Korean populations was examined, providing fundamental data for selective breeding efforts to increase the genetic diversity of Korean *C. gigas*, which could also serve as valuable baseline data for Pacific oyster breeding improvement in other regions.

High genetic diversity within a population enhances its ability to adapt to environmental changes [30,31,32]. In this study, the genetic diversity of 13 wild populations was found to be low [33]. Generally, in wild populations, the observed (*H*_O_) and expected heterozygosity (*H*_E_) are similar unless affected by factors like bottlenecks, founder effects, or human intervention [30,31,32]. However, in this study, all populations showed significantly lower H_O_ than H_E_, likely due to genetic diversity-lowering factors such as bottlenecks, population declines, or inbreeding. HWE is usually stable in wild populations, so deviations across all groups suggest an event disrupting equilibrium. The inbreeding coefficient (*F*_IS_) was significant across all populations, indicating ongoing inbreeding and a potential for genetic diversity loss over time [34,35]. A previous study by An et al. on Korean *C. gigas* (An et al., 2013, *H*_O_ = 0.777, *H*_E_ =0.918) reported high genetic diversity, but *H*_O_ was lower than *H*_E_, indicating low observed genetic diversity [2]. Although *H*_O_ values may differ across markers, the trend of genetic diversity based on the *H*_E_ and *H*_O_ relationship was similar between past and present studies [2].

Bottlenecks are often caused by events such as natural disasters, climate change, or disease that rapidly reduce population size [36]. This reduction diminishes genetic diversity, making it crucial to analyze and manage population genetic diversity [37]. All populations showed evidence of bottlenecks, but the L-shaped distribution indicated that, although bottlenecks were detected previously, the genotype distribution has now normalized, suggesting recovery [38]. While the infinite alleles model (IAM) is suitable for identifying recent bottlenecks, historical bottlenecks are better estimated using the M ratio, where values below 0.6 imply a past bottleneck [39,40]. TPM and SMM models are more conservative in detecting bottlenecks than the IAM model and are less likely to detect bottlenecks once they have recovered [39]. Therefore, there is a possibility that a bottleneck existed when it was detected under the IAM model [40].

In this study, M-ratio values (<0.6) suggest historical bottlenecks in the 13 populations of *C. gigas*, although recovery has since occurred. While the specific causes of these bottlenecks remain unclear, the genetic evidence shows that bottleneck events affected all populations in this study. Species in intertidal zones, like *C. gigas*, experience extreme environmental shifts due to tides, which can lead to population declines [41,42,43]. Given that *C. gigas* inhabits these zones, it is likely that extreme environmental changes contributed to population declines, resulting in bottleneck effects.

Effective population size (*N*_e_) plays a crucial role in minimizing inbreeding, genetic drift, and preserving genetic diversity [15,44]. Larger *N*_e_ slows genetic diversity loss and reduces the impact of inbreeding [43]. Therefore, to enhance the genetic diversity of *C. gigas*, which is the goal of this study, it is essential to select populations with a larger *N*_e_. In this study, GGD, JD, SA, SR, and WSD were identified as populations with *N*_e_ greater than 500. These populations are ideal for breeding, as their genetic diversity is easier to maintain across generations [45,46]. Overall, considering *N*_e_ and genetic diversity, GGD, JD, SA, SR, and WSD are the most suitable for breed improvement. Since these groups have lower *H*_O_ than *H*_E_ and few individuals with rare alleles or heterozygous genotypes, selective crossbreeding may be necessary to introduce more genetically diverse individuals.

In this study, populations such as AD and CSD showed very low *N*_e_, which may be due to the population characteristics of *C. gigas*. Wild populations of *C. gigas* are distributed in the intertidal zone but are not widely distributed. In the field survey for sampling, it was confirmed that *C. gigas* were concentrated in certain intertidal zones where they had ample attachment and feeding spots. This concentrated population structure may have affected *N*_e_ when the initial population was formed. Since *N*_e_ is affected by the genetic diversity of the parent population, if the larvae were fertilized from a small number of parent populations when the wild AD and CSD populations were formed or if there was continuous inbreeding, this may be the reason for the low *N*_e_.

### 4.2. Genetic Structure of Wild Populations for C. gigas

Genetic structure also plays a key role in enhancing genetic diversity [47]. It provides reference data to guide the selection of populations for breeding programs [48]. In this study, the genetic differentiation index (*F*_ST_) values were quite low across all groups (*F*_ST_ = 0.001–0.057), indicating very limited genetic differentiation. This level of differentiation is not consistent with species-level divergence [32]. Therefore, it appears that all populations could serve as suitable reference populations for promoting genetic diversity.

To further investigate the genetic structure, we used STRUCTURE software, which identified two clusters: one consisting of UD alone and the other comprising the remaining populations (AD, CC, CSD, GGD, JD, MA, SA, SJ, SR, WD, WSD, and YD) at *K* = 2. However, because STRUCTURE cannot recognize a single group as such, we employed DAPC for a more detailed analysis. As a non-model estimation method that visually represents group differences in a scatter plot, DAPC showed that all groups clustered together, effectively forming a single genetic group. This finding indicates that the Korean *C. gigas* population represents a single, unified genetic structure.

In summary, considering both genetic diversity and structure, we selected the GGD, JD, SA, SR, and WSD populations as sources to enhance the genetic diversity of Korean *C. gigas*. Given the low genetic differentiation, blending these populations is unlikely to cause significant genetic confusion. Consequently, GGD, JD, SA, SR, and WSD stand out as appropriate reference populations for selective breeding aimed at increasing the genetic diversity of Korean *C. gigas*.

## 5. Conclusions

This study examined the genetic diversity and structure of 13 wild populations of *Crassostrea gigas* in Korea. Overall, the genetic diversity of Korean *C. gigas* was relatively low, with observed heterozygosity (*H*_O_) ranging from 0.367 to 0.418 and expected heterozygosity (*H*_E_) from 0.580 to 0.661. Effective population sizes varied widely, from 28 to 2491. Genetic structure analysis revealed that all populations appeared as one group. This study identified a reference population to guide the selection of a breeding population aimed at enhancing genetic diversity in Korean *C. gigas*, providing foundational data on genetic diversity and structure that can support future selective breeding efforts in *C. gigas*.

## Figures and Tables

**Figure 1 biology-14-00449-f001:**
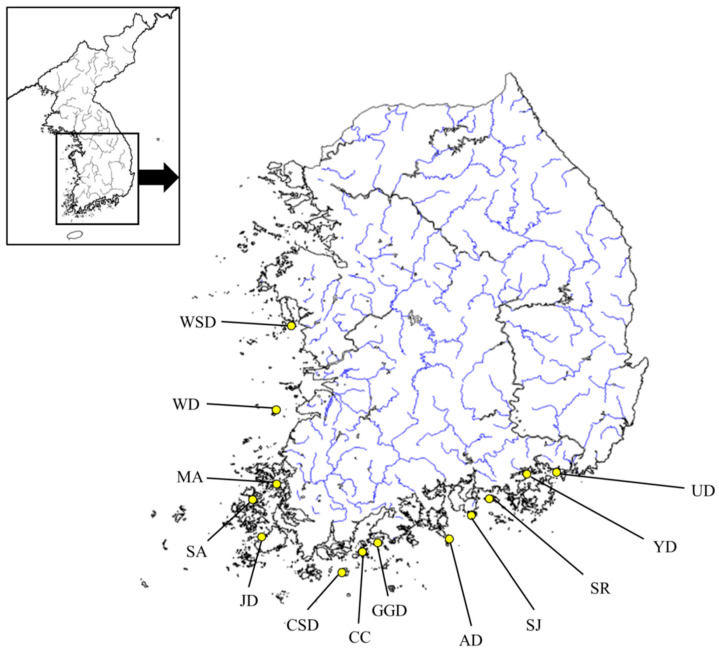
Sampling sites for the thirteen populations of *C. gigas*, with population abbreviations listed in Table 1. The blue line is the stream of the river.

**Figure 2 biology-14-00449-f002:**
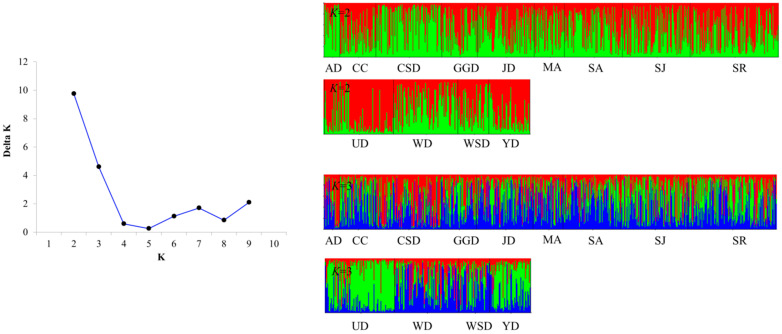
Genetic structure of *C. gigas* populations (*K* = 2, 3), showing delta *K* values for population. Each histogram illustrates the probability of an individual, represented by a specific color, being assigned to a given cluster.

**Figure 3 biology-14-00449-f003:**
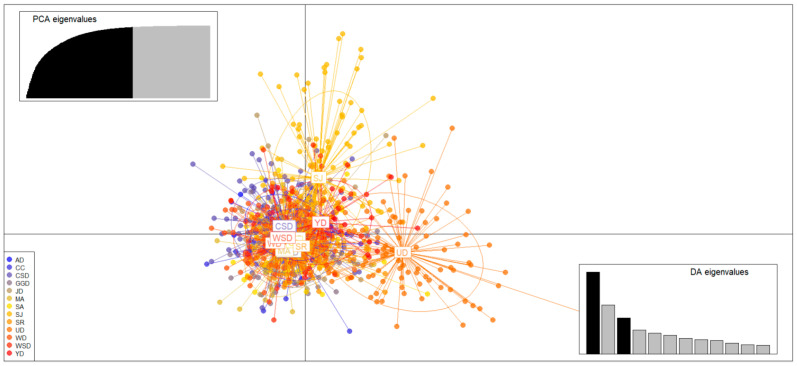
Scatterplots from the discriminant analysis of principal components (DAPC) for *C. gigas* are shown. Each color represents a unique population, indicating separate genetic clusters, with population abbreviations labeled for each cluster. The top-left graph displays the contribution of eigenvalues from the selected principal components, while the bottom-right graph illustrates the variance explained by the eigenvalues of the two discriminant functions in the scatterplot.

**Table 1 biology-14-00449-t001:** Location details for the thirteen wild populations of *C. gigas*.

Code	Location Name	Years	*N*	Location
AD	Ando	2024	26	34°29′12″ N 127°48′17″ E
CC	Chukchi	2024	60	34°22′14″ N 127°02′04″ E
CSD	Chungsando	2024	109	34°10′48″ N 126°51′23″ E
GGD	Gugeumdo	2024	59	34°29′13″ N 127°07′25″ E
JD	Jindo	2024	94	34°25′18″ N 126°20′43″ E
MA	Muan	2024	50	34°56′21″ N 126°23′25″ E
NDD	Nodaedo	2024	25	34°40′09″ N 128°14′47″ E
SA	Sinan	2024	96	34°51′58″ N 126°18′53″ E
SJ	Sangju	2024	112	34°42′48″ N 127°59′12″ E
SR	Saryang	2024	146	34°50′00″ N 128°08′13″ E
UD	Udo	2024	89	35°05′16″ N 128°43′23″ E
WD	Wido	2024	83	35°35′47″ N 126°16′04″ E
WSD	Wonsando	2024	40	36°22′03″ N 126°23′42″ E
YD	Yangdo	2024	53	35°04′55″ N 128°29′30″ E

*N*: number of samples.

**Table 2 biology-14-00449-t002:** Genetic diversity summary for 13 populations of *C. gigas* based on 15 microsatellite loci.

ID	Location	*N*	*N_A_*	*H* _O_	*H* _E_	*P* _HWE_	*F* _IS_
AD	AnDo	26	7.60	0.385	0.628	0.000 ***	0.393 ***
CC	ChukChi	60	9.60	0.404	0.593	0.000 ***	0.318 ***
CSD	ChungSanDo	109	11.07	0.406	0.600	0.000 ***	0.319 ***
GGD	GuGeumDo	59	9.13	0.393	0.585	0.000 ***	0.325 ***
JD	JinDo	94	10.60	0.339	0.580	0.000 ***	0.417 ***
MA	MuAn	50	9.00	0.397	0.605	0.000 ***	0.345 ***
SA	SinAn	96	10.47	0.418	0.593	0.000 ***	0.297 ***
SJ	SangJu	112	12.27	0.398	0.661	0.000 ***	0.399 ***
SR	SaRyang	146	12.07	0.412	0.596	0.000 ***	0.310 ***
UD	UDo	89	10.13	0.367	0.614	0.000 ***	0.403 ***
WD	WiDo	83	10.40	0.406	0.600	0.000 ***	0.324 ***
WSD	WonSanDo	40	8.60	0.395	0.592	0.000 ***	0.335 ***
YD	YangDo	53	9.00	0.413	0.591	0.000 ***	0.301 ***

*N*: number of samples; *N*_A_: number of average alleles; *H*_O_: observed heterozygosity; *H*_E_: expected heterozygosity; *P*_HWE_: Hardy–Weinberg equilibrium value; *F*_IS_: inbreeding coefficient. *** *p* < 0.001.

**Table 3 biology-14-00449-t003:** Summary of bottleneck, effective population sizes, and M-ratio in thirteen populations.

Population ID	*N*	Wilcoxon Sign-Rank Test				M Ratio	*N* _e_	(95% CI)
		*P* _IAM_	*P* _TPM_	*P* _SMM_	Mode-Shift			
AD	26	0.000 ***	0.268	0.542	L-shaped	0.333	28	(21–37)
CC	60	0.003 **	0.890	0.389	L-shaped	0.315	273	(155–918)
CSD	109	0.001 **	0.762	0.890	L-shaped	0.319	90	(76–110)
GGD	59	0.001 **	0.934	0.421	L-shaped	0.335	897	(263–∞)
JD	94	0.001 **	0.890	0.489	L-shaped	0.349	2491	(539–∞)
MA	50	0.000 ***	0.389	0.978	L-shaped	0.333	388	(183–∞)
SA	96	0.000 ***	0.542	0.542	L-shaped	0.367	1709	(501–∞)
SJ	112	0.002 **	0.978	0.303	L-shaped	0.374	441	(295–823)
SR	146	0.002 **	0.330	0.035	L-shaped	0.326	1381	(607–∞)
UD	89	0.002 **	0.561	0.890	L-shaped	0.358	314	(206–610)
WD	83	0.010 *	0.890	0.229	L-shaped	0.330	-	(2364–∞)
WSD	40	0.000 ***	0.358	0.761	L-shaped	0.357	1489	(434–∞)
YD	53	0.000 ***	0.229	0.389	L-shaped	0.335	-	(1210–∞)

*N*: number of samples; *P*_IAM_: *p*-value of bottleneck test using infinite allele mutation model; *P*_TPM_: *p*-value of bottleneck test using two-phase mutation model (10% variance and 90% proportions of SMM); *P*_SMM_: *p*-value of bottleneck test using stepwise mutation model; *N*_e_: estimated effective population size by NeEstimator software; CI: confidence interval. * *p* < 0.05; ** *p* < 0.01 *** *p* < 0.001.

**Table 4 biology-14-00449-t004:** *F*_ST_ among populations by microsatellite loci data of *C. gigas*.

	AD	CC	CSD	GGD	JD	MA	SA	SJ	SR	UD	WD	WSD	YD
AD	-	0.002	0.152	0.014	0.000	0.002	0.000	0.000	0.000	0.000	0.008	0.006	0.000
CC	0.021 *	-	0.000	0.618	0.646	0.392	0.797	0.000	0.654	0.000	0.406	0.557	0.015
CSD	0.008	0.018 *	-	0.000	0.000	0.000	0.000	0.000	0.000	0.000	0.000	0.000	0.000
GGD	0.016 *	0.002	0.015 *	-	0.002	0.720	0.290	0.000	0.496	0.000	0.504	0.249	0.000
JD	0.023 *	0.002	0.017 *	0.009 *	-	0.004	0.536	0.000	0.009	0.000	0.014	0.266	0.007
MA	0.020 **	0.004	0.016 *	0.002	0.009 *	-	0.285	0.000	0.020	0.000	0.055	0.293	0.000
SA	0.020 *	0.001	0.013 *	0.003	0.002	0.003	-	0.000	0.772	0.000	0.254	0.819	0.000
SJ	0.020 *	0.013 *	0.018 *	0.015 *	0.012 *	0.015 *	0.012 *	-	0.000	0.000	0.000	0.000	0.000
SR	0.018 *	0.001	0.013 *	0.002 *	0.004 *	0.006 *	0.000	0.012 *	-	0.000	0.080	0.104	0.001
UD	0.057 *	0.027 *	0.047 *	0.033 *	0.036 *	0.029 *	0.029 *	0.029 *	0.029 *	-	0.000	0.000	0.000
WD	0.015 *	0.003	0.015 *	0.002 *	0.006 *	0.006	0.002	0.013 *	0.003	0.041 *	-	0.381	0.000
WSD	0.021 *	0.003	0.017 *	0.005	0.005	0.005	0.001	0.014 *	0.005	0.035 *	0.003	-	0.000
YD	0.034 *	0.009 *	0.026 *	0.019 *	0.009 *	0.021 *	0.013 *	0.014 *	0.009 *	0.026 *	0.020 *	0.020 *	-

Pairwise genetic differentiation significance level (above). *F*_st_: pairwise genetic differentiation (below); * values less than 0.05 are significant. ** values less than 0.01 are significant.

**Table 5 biology-14-00449-t005:** Detailed summary of AMOVA (analysis of molecular variance) findings for wild populations.

Source of Variation	d.f.	Sum of Squares	Variance Components	Percentage of Variance	*F*-Statistics
Microsatellite loci (One group based on the DAPC: AD, CC, CSD, GGD, JD, MA, SA, SJ, SR, UD, WD, WSD, and YD)					
Among groups	12	176.028	0.05567	1.22	0.012 ***
Among populations within groups	1004	6108.802	1.56091	34.09	0.345 ***
Within populations	1017	3013.000	2.96264	64.70	0.353 ***
Total	2033	9297.830	4.57922	100.00	-

d.f.: degrees of freedom. *** *p* < 0.001. *F*-statistics are based on standard permutation across the full dataset.

## Data Availability

The raw data supporting the conclusions of this article will be made available by the authors, without undue reservation.

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
