# Peer review of "Genetic Diversity and Structure of Korean Pacific Oyster (Crassostrea gigas) for Determining Selective Breeding Groups"

_biology, 2025, doi:10.3390/biology14040449_

Round 1

Reviewer 1 Report

Comments and Suggestions for Authors

 General Assessment

This study provides a comprehensive analysis of the genetic diversity and structure of 13 wild populations of *Crassostrea gigas* in Korea. The research is well-designed, methodologically sound, and addresses an important issue in aquaculture—maintaining and enhancing genetic diversity for sustainable breeding programs. The findings are significant for the Korean Pacific oyster industry and could serve as a model for similar studies in other regions. Below are specific comments and suggestions for improvement.

---

 Strengths

  1. Clear Objectives: The study aims to assess genetic diversity and structure to identify reference populations for selective breeding, which is clearly articulated.
  2. Robust Methodology: The use of microsatellite markers, bottleneck tests, effective population size estimation, and genetic structure analyses (STRUCTURE and DAPC) is appropriate and well-executed.
  3. Comprehensive Data: The study includes a large number of samples (13 populations) and multiple genetic metrics (e.g., heterozygosity, inbreeding coefficients, FST values), providing a thorough dataset.
  4. Practical Implications: The identification of populations with high effective population sizes (e.g., GGD, JD, SA, SR, WSD) for selective breeding is valuable for aquaculture management.
  5. Well-Structured Discussion: The discussion contextualizes the findings within broader literature and highlights the implications for conservation and breeding programs.

---

 Areas for Improvement

# 1. Clarity and Flow

- Introduction: The introduction could be streamlined to better highlight the knowledge gap and the study's novelty. For example, the transition from general oyster industry issues to the specific need for genetic diversity assessment could be smoother.

- Results: Some sections are dense with numbers (e.g., Table 2). Consider summarizing key trends in the text and relegating detailed values to tables or supplementary materials.

# 2. Statistical and Technical Details

- Bottleneck Tests: The manuscript mentions that all populations showed bottleneck effects under IAM but not under TPM/SMM. This discrepancy should be discussed further—why might IAM detect bottlenecks while TPM/SMM do not? Is this due to model assumptions or biological reasons?

- Effective Population Size (Ne): The Ne estimates vary widely (28–2491). The discussion should address why some populations (e.g., AD) have very low Ne and whether this affects their suitability as reference populations.

- Genetic Structure: The STRUCTURE analysis suggests K=2 (UD vs. others), while DAPC supports a single group. This discrepancy should be clarified. Is UD truly distinct, or is this an artifact of the analysis?

# 3. Interpretation and Context

- Low Genetic Diversity: The study reports low genetic diversity compared to past studies (e.g., An et al., 2013). The authors should discuss potential reasons for this decline (e.g., overharvesting, habitat loss, or methodological differences).

- Historical Bottlenecks: The M-ratio results suggest historical bottlenecks, but the causes are speculative (e.g., environmental shifts). Are there historical records (e.g., climate events, disease outbreaks) that could support this?

- Implications for Breeding: The manuscript recommends using GGD, JD, SA, SR, and WSD for breeding due to their high Ne. However, it does not address how to mitigate the low diversity in other populations. Could translocation or assisted gene flow be considered?

# 4. Presentation and Figures

- Figure Quality: The figures (e.g., STRUCTURE bar plots, DAPC scatterplots) are described but not included in the provided text. Ensure these are high-resolution and clearly labeled in the final version.

- Table Readability: Table 4 (FST values) is hard to parse due to the dual presentation of significance and values. Consider splitting this into two tables or using footnotes for significance.

# 5. Minor Corrections

- Typos and Grammar:

  - Page 1: "de-creasing" should be "decreasing."

  - Page 2: "There‑fore" should be "Therefore."

  - Page 9: "breed improvement" could be "breeding improvement."

- Citations: Some references are incomplete (e.g., "An et al., 2013" is cited before the full reference appears in the list). Ensure all in-text citations match the reference list.

---

 Conclusion

This manuscript presents important findings on the genetic diversity and structure of *Crassostrea gigas* in Korea, with clear applications for selective breeding programs. With minor revisions to improve clarity, statistical interpretation, and contextualization, the study will be a valuable contribution to the field. Below is a summary of recommendations:

  1. Clarify discrepancies between STRUCTURE and DAPC results.
  2. Expand discussion on causes of low diversity and bottleneck effects.
  3. Improve readability of tables and figures.
  4. Address minor grammatical and typographical errors.

Overall, this is a strong manuscript that, with these revisions, will be suitable for publication.

Recommendation: Accept with Minor Revisions

Author Response

Comments 1:

Strengths

  1. Clear Objectives: The study aims to assess genetic diversity and structure to identify reference populations for selective breeding, which is clearly articulated.
  2. Robust Methodology: The use of microsatellite markers, bottleneck tests, effective population size estimation, and genetic structure analyses (STRUCTURE and DAPC) is appropriate and well-executed.
  3. Comprehensive Data: The study includes a large number of samples (13 populations) and multiple genetic metrics (e.g., heterozygosity, inbreeding coefficients, FST values), providing a thorough dataset.
  4. Practical Implications: The identification of populations with high effective population sizes (e.g., GGD, JD, SA, SR, WSD) for selective breeding is valuable for aquaculture management.
  5. Well-Structured Discussion: The discussion contextualizes the findings within broader literature and highlights the implications for conservation and breeding programs.

Response 1: Thank you for pointing this out. We investigated the genetic diversity underlying the selection of a base population for selective breeding of C. gigas. We believe that the comprehensive sampling population size and sufficient sampling number provide robust data.

Comments 2: # 1. Clarity and Flow

- Introduction: The introduction could be streamlined to better highlight the knowledge gap and the study's novelty. For example, the transition from general oyster industry issues to the specific need for genetic diversity assessment could be smoother.

- Results: Some sections are dense with numbers (e.g., Table 2). Consider summarizing key trends in the text and relegating detailed values to tables or supplementary materials.

Response 2: Thank you for your review. We have revised this manuscript to emphasize the need for genetic diversity.

[Crassostrea gigas (Pacific oyster) is a key species in the Korean aquaculture industry [1]. In recent years, however, the production of farmed C. gigas has shown a marked decline, raising concerns about the long-term sustainability of oyster farming. One of the primary factors contributing to this decline is the reduction in genetic diversity within cultured populations [2,3]. A decrease in genetic diversity has been reported to correlate with lower early survival rates, particularly during the larval stages [4]. As a result, significant efforts have been made to develop strategies aimed at preserving and enhancing genetic variation in aquaculture stocks [3,4].

Among these strategies, selective breeding approaches that explicitly consider both the number and diversity of alleles are recognized as crucial for restoring and maintaining genetic diversity [5]. Increasing the frequency of heterozygous individuals carrying diverse allelic combinations can improve the adaptive potential and overall genetic health of oyster populations. Therefore, the development of well-structured breeding designs and targeted selection programs is essential for enhancing genetic diversity and ensuring the long-term viability of the Pacific oyster aquaculture industry [6]. Meanwhile, the genetic diversity of wild Pacific oysters from Korea has been shown to be high [3].]

Comments 3:

- Results: Some sections are dense with numbers (e.g., Table 2). Consider summarizing key trends in the text and relegating detailed values to tables or supplementary materials.

Response 3: Thank you for your review. Please understand that it is difficult to summarize the important values ​​because the values ​​of all indicators in Table 2 must be considered as a whole.

Comments 4:

# 2. Statistical and Technical Details

- Bottleneck Tests: The manuscript mentions that all populations showed bottleneck effects under IAM but not under TPM/SMM. This discrepancy should be discussed further—why might IAM detect bottlenecks while TPM/SMM do not? Is this due to model assumptions or biological reasons?

- Effective Population Size (Ne): The Ne estimates vary widely (28–2491). The discussion should address why some populations (e.g., AD) have very low Ne and whether this affects their suitability as reference populations.

- Genetic Structure: The STRUCTURE analysis suggests K=2 (UD vs. others), while DAPC supports a single group. This discrepancy should be clarified. Is UD truly distinct, or is this an artifact of the analysis?

Response 4: Thank you for your review. The comments have been revised and reflected in the manuscript.

*[TPM and SMM models are more conservative in detecting bottlenecks than the IAM model and are less likely to detect bottlenecks once they have recovered [39]. Therefore, there is a possibility that a bottleneck existed when it was detected under the IAM model [40].]:

*[In this study, populations such as AD and CSD showed very low Ne, which may be due to the population characteristics of C. gigas. Wild populations of C. gigas are distributed in the intertidal zone, but are not widely distributed. In the field survey for sampling, it was confirmed that C. gigas were concentrated in certain intertidal zones where they were good for attachment and feeding spots. This concentrated population structure may have affected Ne when the initial population was formed. Since Ne is affected by the genetic diversity of the parent population, if the larvae were fertilized from a small number of parent populations when the wild AD and CSD populations were formed, or if there was continuous inbreeding, this may be the reason for the low Ne.]

*Due to the nature of the STRUCTURE analysis program, the delta K value cannot be obtained when K = 1. Therefore, if the histogram distribution shows a panmixia shape when K = 2, it is likely that K = 1, and this can be more clearly identified through the DAPC method.

Comments 5:

# 3. Interpretation and Context

- Low Genetic Diversity: The study reports low genetic diversity compared to past studies (e.g., An et al., 2013). The authors should discuss potential reasons for this decline (e.g., overharvesting, habitat loss, or methodological differences).

- Historical Bottlenecks: The M-ratio results suggest historical bottlenecks, but the causes are speculative (e.g., environmental shifts). Are there historical records (e.g., climate events, disease outbreaks) that could support this?

- Implications for Breeding: The manuscript recommends using GGD, JD, SA, SR, and WSD for breeding due to their high Ne. However, it does not address how to mitigate the low diversity in other populations. Could translocation or assisted gene flow be considered?

Response 5:

Thank you for your review. The difference in genetic diversity was mentioned in the manuscript as follows.

*It was mentioned that HO may differ due to marker differences, but the decreasing trend is similar when looking at the difference between HE and HO.

[Although HO values may differ across markers, the trend of genetic diversity based on the HE and HO relationship was similar between past and present studies [2].]

* In Korea, there are records that can be used to infer climate change through records from the 500 years of the Joseon Dynasty, but since the current research cannot determine the point in time when the bottleneck occurred, we can only guess. We preparing a paper that studies the historical Ne changes through subsequent genomes, so I think that the paper prepared in the future will be able to answer your questions when it is published. Thank you for your understanding.

*The way to increase Ne can be solved by crossbreeding between genetically distant individuals. However, this crossbreeding study is not the main purpose of this study, so it is considered a problem to be solved in future studies. Thank you for your understanding.

Comments 6:

# 4. Presentation and Figures

- Figure Quality: The figures (e.g., STRUCTURE bar plots, DAPC scatterplots) are described but not included in the provided text. Ensure these are high-resolution and clearly labeled in the final version.

- Table Readability: Table 4 (FST values) is hard to parse due to the dual presentation of significance and values. Consider splitting this into two tables or using footnotes for significance.

Response 6:

Thank you for your review.

*Putting a description on the picture itself would have caused some complexity, so it was added to the picture caption.

* FST added *footnotes to emphasize significance.

Comments 7: # 5. Minor Corrections

- Typos and Grammar:

  - Page 1: "de-creasing" should be "decreasing."

  - Page 2: "There‑fore" should be "Therefore."

  - Page 9: "breed improvement" could be "breeding improvement."

- Citations: Some references are incomplete (e.g., "An et al., 2013" is cited before the full reference appears in the list). Ensure all in-text citations match the reference list.

Response 7:

Thank you for your review.

Minor grammar and reference citation corrections.

Comments 8: Conclusion

This manuscript presents important findings on the genetic diversity and structure of *Crassostrea gigas* in Korea, with clear applications for selective breeding programs. With minor revisions to improve clarity, statistical interpretation, and contextualization, the study will be a valuable contribution to the field. Below is a summary of recommendations:

  1. Clarify discrepancies between STRUCTURE and DAPC results.
  2. Expand discussion on causes of low diversity and bottleneck effects.
  3. Improve readability of tables and figures.
  4. Address minor grammatical and typographical errors.

Response 8:

Thank you for your review.

Clarified discrepancies between structural analysis results and DAPC results.

Expanded discussion on causes of low diversity and bottlenecks.

Improved readability of tables and figures.

Fixed minor grammar and typos.

Reviewer 2 Report

Comments and Suggestions for Authors

In the study, the authors collected a large number of Pacific oyster samples from 13 regions in Korea to study the genetic diversity and structure. This is a relatively complete study, but there are still some questions that need to be answered by the authors.

1.Does South Korea now have artificial breeding facilities for Pacific oysters, and does this have an impact on the genetic diversity of wild Pacific oysters.

2.Enumerate the challenges faced by the Korean oyster farming industry and the possibilities to address them through intraspecific hybridization.

3.The authors need to discuss the findings of the study. In figure 3, I found that the regions located on the east coastline of Korea (UD, YD, SJ) were separated from other regions. Please discusses this phenomenon, if possible.

4.In this manuscript, I’m wondering how to determine selective breeding groups of Korean Pacific Oyster.  

Additionally, here are some sentences that need to be revised.

1.An increase in individuals with various alleles and heterozygotes can improve genetic diversity, but selective crossbreeding is required for this crossbreeding [6].

2.Species in intertidal zones, like C. gigas, experience extreme environmental shifts due to tides, which can lead to population declines [41-43]. 

Comments on the Quality of English Language

The English could be improved to more clearly express the research.

Author Response

For research article

Response to Reviewer 2 Comments

1. Point-by-point response to Comments and Suggestions for Authors

Comments 1:

1.Does South Korea now have artificial breeding facilities for Pacific oysters, and does this have an impact on the genetic diversity of wild Pacific oysters.

Response 1: Thank you for pointing this out.

Currently, we are producing farmed oysters in Korea. The samples collected in this study were collected from wild Pacific oysters far from the farm. Since the distance that farmed oysters can spread is limited, we believe that they will not have a significant impact on genetic diversity.

Comments 2:

2.Enumerate the challenges faced by the Korean oyster farming industry and the possibilities to address them through intraspecific hybridization.

Response 2: Thank you for your review.

This study presents the basis for genetic diversity to be selected as a base population, so the possibility of resolving the problem through crossbreeding, which will be studied in the future, will be discussed in a paper prepared next time. We ask for your understanding.

Comments 3:

3.The authors need to discuss the findings of the study. In figure 3, I found that the regions located on the east coastline of Korea (UD, YD, SJ) were separated from other regions. Please discusses this phenomenon, if possible.

Response 3: Thank you for your review.

In the case of STRUCTURE, the Panmixia population is not detected. Therefore, since the results clearly detected in DAPC are explained as a single population, it cannot be considered that a clear separation has occurred. We ask for your understanding.

Comments 4:

4.In this manuscript, I’m wondering how to determine selective breeding groups of Korean Pacific Oyster.  

Response 4: Thank you for your review.

Considering genetic diversity (HO), since there is no population larger than HE, we mainly selected populations in order of large Ne. We ask for your understanding.

Comments 5:

1.An increase in individuals with various alleles and heterozygotes can improve genetic diversity, but selective crossbreeding is required for this crossbreeding [6].

Response 5:

Thank you for your review.

This part was revised by reviewer 1 and was not re-edited as a revised sentence. Please understand.

Comments 6:

2.Species in intertidal zones, like C. gigas, experience extreme environmental shifts due to tides, which can lead to population declines [41-43].

Response 6:

Thank you for your review.

This sentence has been revised.

Round 2

Reviewer 2 Report

Comments and Suggestions for Authors

It should be accepted for publication.